# Orbital textures and evolution of correlated insulating state in monolayer 1T phase transition metal dichalcogenides

Qiang Gao[1,5], Haiyang Chen[1,5], Wen-shin Lu[2,3], Yang-hao Chan[2,3], Zhenhua Chen[4], Yaobo Huang [4], Zhengtai Liu[4] & Peng Chen [1 ✉]

Strong electron-electron interaction can induce Mott insulating state, which is believed to host unusual correlated phenomena such as quantum spin liquid when quantum fluctuation dominates and unconventional superconductivity through doping. Transition metal compounds as correlated materials provide a versatile platform to engineer the Mott insulating state. Previous studies mostly focused on the controlling of the repulsive interaction and bandwidth of the electrons by gating or doping. Here, we performed angle-resolved photo-emission spectroscopy (ARPES) on monolayer 1T phase $NbSe_2$, $TaSe_2$, and $TaS_2$ and directly observed their band structures with characteristic lower Hubbard bands. By systematically investigating the orbital textures and temperature dependence of the energy gap of the materials in this family, we discovered that hybridization of the chalcogen $p$ states with lower Hubbard band stabilizes the Mott phase via tuning of the bandwidth, as shown by a significant increase of the transition temperature ($T_C$) at a stronger hybridization strength. Our findings reveal a mechanism for realizing a robust Mott insulating phase and establish monolayer 1T phase transition metal dichalcogenide family as a promising platform for exploring correlated electron problems.

When repulsive Coulomb interaction is strong that conducting electrons become localized in system with half-filled bands, energy gap will be formed, leading to the Mott insulating ground state[1-5]. In a simplified Hubbard model, the existence of Mott phase is determined by the relative strength of Coulomb interaction ($U$) and the width of the half-filled band ($W$)[1]. Tuning of the interaction strength ($U/W$) can effectively drive the Mott transition in moiré superlattice via gating or in natural materials by doping[6-11]. Transition metal compounds with unpaired $d$ electrons in transition metal ions can exhibit Mott insulating states[1,2], which are characterized by an opening of energy gap and emergence of flat bands known as Hubbard bands close to the Fermi level in the low temperature. Understanding of the Hubbard band structure is thus essential to the Mott physics.

$NbSe_2$, $TaSe_2$, and $TaS_2$ are members of the broad class of 1T phase layered triangular lattice materials. A first-order phase transition to a commensurate charge density wave (CDW) state has been reported in this family of materials[12-16]. As illustrated in Fig. 1a, the 1T phase $MX_2$ (M = Ta, Nb; X = Se, S) family has the same trigonal crystal symmetry but slightly different lattice constants. In a simple ionic picture, the chalcogen atoms form a distorted octahedron around the transition metal atom. The atomic $d$ levels split, and the energy of $d_{3z^2-r^2}$ orbital is lowered. The nominal $d^1$ valence electron results in half-filled $d_{3z^2-r^2}$ state and leads to a metallic band structure. In the low temperature, the in-plane atoms deformed into Star-of-David (SOD) clusters with a $(\sqrt{13} \times \sqrt{13})$ periodicity[17] (Fig. 1a). In a SOD unit cell, twelve in-plane Ta/Nb atoms distorted towards the central atom and twelve electrons

[1]Key Laboratory of Artificial Structures and Quantum Control (Ministry of Education), Tsung-Dao Lee Institute, Shanghai Center for Complex Physics, School of Physics and Astronomy, Shanghai Jiao Tong University, Shanghai, China. [2]Institute of Atomic and Molecular Sciences, Academia Sinica, Taipei, Taiwan. [3]Physics Division, National Center for Theoretical Sciences, Taipei, Taiwan. [4]Shanghai Synchrotron Radiation Facility, Shanghai Advanced Research Institute, Chinese Academy of Sciences, Shanghai, China. [5]These authors contributed equally: Qiang Gao, Haiyang Chen. ✉e-mail: pchen229@sjtu.edu.cn

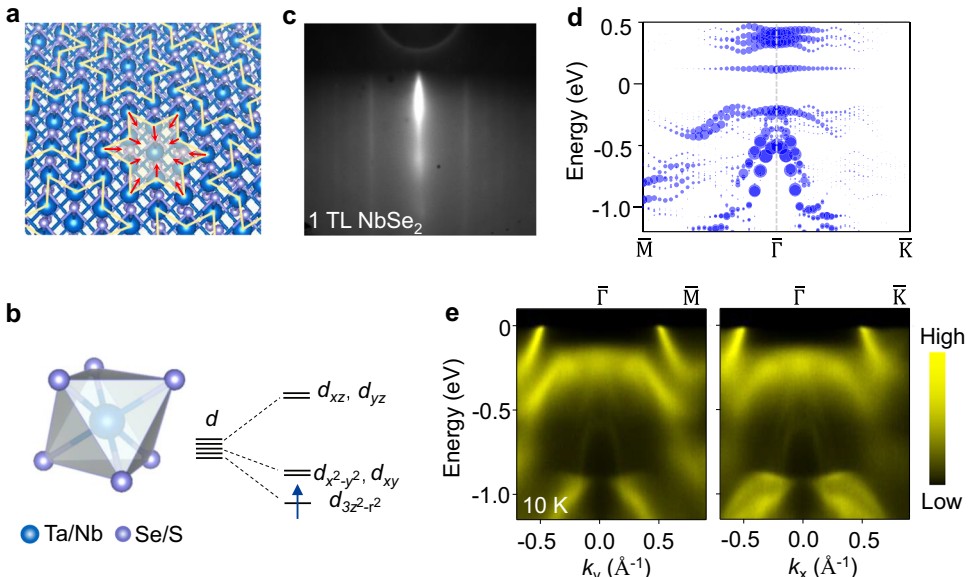

**Fig. 1 | Structure and electronic bands of monolayer NbSe₂ films. a** Schematic of the triangular lattice and displacements of metal atoms in the Star-of-David CDW cluster. **b** The $d$ orbitals splitting in the distorted octahedral crystal fields. **c** A RHEED pattern of a monolayer NbSe₂ film. **d** Unfolded band dispersion for monolayer 1T-NbSe₂ with the ($\sqrt{13} \times \sqrt{13}$) superstructure along the $\overline{M\Gamma K}$ direction. **e** ARPES spectra taken along the $\overline{\Gamma M}$ and $\overline{\Gamma K}$ directions for monolayer NbSe₂ with 40 eV $p$-polarized light.

with one from each transition metal atom occupy six full-filled bands. One unpaired electron localized on the central Ta atom forms a half-filled band, which is predicted to cross the Fermi level from the conventional band theory. Previous experiments have shown that monolayer 1T-MX₂ (M = Ta, Nb; X = Se, S) systems are insulating in commensurate CDW state with Star of David superstructure[18–22]. Mott-Hubbard mechanism was used to explain the ground state, but the Hubbard band fine structure has not been revealed, and the effect of coupling with chalcogen orbitals remains elusive.

Here, we study the electronic structure of monolayer 1T-NbSe₂, 1T-TaSe₂, and 1T-TaS₂ by ARPES and show evolution of the Mott state with varying hybridization strength between ligand and transition metal states. Low dimensionality simplifies the interlayer interaction problem from various stacking orders[23,24] and reduces the screening around the Fermi level, which in principle increases the Coulomb interaction and leads to a more robust Mott state compared to the bulk counterpart[25–28]. Sharp diffraction patterns measured by reflection high energy electron diffraction (RHEED) indicates a well-ordered monolayer NbSe₂ grown by molecular beam epitaxy (MBE), as shown in Fig. 1c. ARPES spectra of monolayer NbSe₂ along $\overline{\Gamma M}$ and $\overline{\Gamma K}$ direction are shown in Fig. 1e. A nearly flat band at ~0.22 eV around the $\overline{\Gamma}$ point is observed[21,22,29,30]. The hole-like Se 4$p$ bands penetrate the flat band and exhibit a continuum structure with an inverted triangle shape. The diminished spectral intensity between the flat band and the Fermi level indicates a gap opening and the system is insulating in the low temperature. Note that the bands crossing the Fermi level at ~±0.5 Å⁻¹ are from the 1H-NbSe₂ as the 1T and 1H phase monolayer NbSe₂ coexist in the films. These metallic bands can be used as a reference to roughly confirm the position of the Fermi level. Dispersive Nb 4$d$ band is evidenced at a larger momentum. No obvious change in these bands except the intensity with different photon energies (matrix element effect), in agreement with the 2D nature of the electronic structure of the system (Supplementary Figs. 3 and 4).

## Results
### Electronic band structure
High energy-resolution ARPES spectra of 1T phase monolayer NbSe₂, TaSe₂, and TaS₂ are shown in Fig. 2. The overall electronic structure

including the band gap and flat band structures are similar in these materials and in excellent agreement with the calculated unfolded band dispersions (Fig. 2c). A ferromagnetic ground state and Hubbard U are included in the calculations in Fig. 2c to reproduce the experimental results. The location of lower Hubbard band (LHB) is determined by the $d_{3z^2-r^2}$ orbital projection from the spectra weight of the central transition metal atom in the $\sqrt{13} \times \sqrt{13}$ superstructure. The difference between the band structure of 1T phase NbSe₂ and TaSe₂ is mainly quantitative that Se bands are closer to the Fermi level, so that there is a clear overlap (~37 meV) between the top Se 4$p$ band and the LHB in monolayer NbSe₂. Such an overlap is absent in TaS₂ as the S 3$p$ bands are in higher binding energies. In the $\sqrt{13} \times \sqrt{13}$ superstructure unfolded results (Fig. 2c), the LHB of these materials exhibits a dip structure around the $\overline{\Gamma}$ point with the maximum depth observed in TaS₂. ARPES spectra of monolayer TaS₂ exhibit a nearly dispersionless flat band, which is rationalized as the LHB fractionalized into a continuum structure because of the strong quantum fluctuations[4,31,32]. In NbSe₂ and TaSe₂, the dip is still visible in ARPES results, and the structure exhibits an inverted triangle shape as the Se 4$p$ states hybridize with the LHB.

### Temperature dependence of the energy gaps
We investigate the insulating ground state of 1T phase monolayer 1T-MX₂ (M = Ta, Nb; X = Se, S) by performing temperature-dependent scans. As shown in Fig. 3, the flat band and the CDW folded bands become blurry with increasing temperature in monolayer NbSe₂. They are almost indiscernible at 330 K, which makes it difficult to extract the correct energy gap by fitting to a phenomenological function[33]. We symmetrized the energy distribution curves (EDCs) at the $\overline{\Gamma}$ point with respect to the Fermi level to illustrate the gap formation (Fig. 3d). The symmetrized EDC at 10 K shows a U-shaped structure around the Fermi level, indicating opening of a gap. At higher temperatures, a V-shaped structure is developed, demonstrating there are some spectral intensities filling up the gap, which is diminishing with increasing temperature. We extract the gap value using the leading-edge midpoint (LEM) of the EDCs. The square of the energy gap as a function of temperature follows a functional form described by a semi-phenomenological mean-field equation (blue solid curve in Fig. 3e).

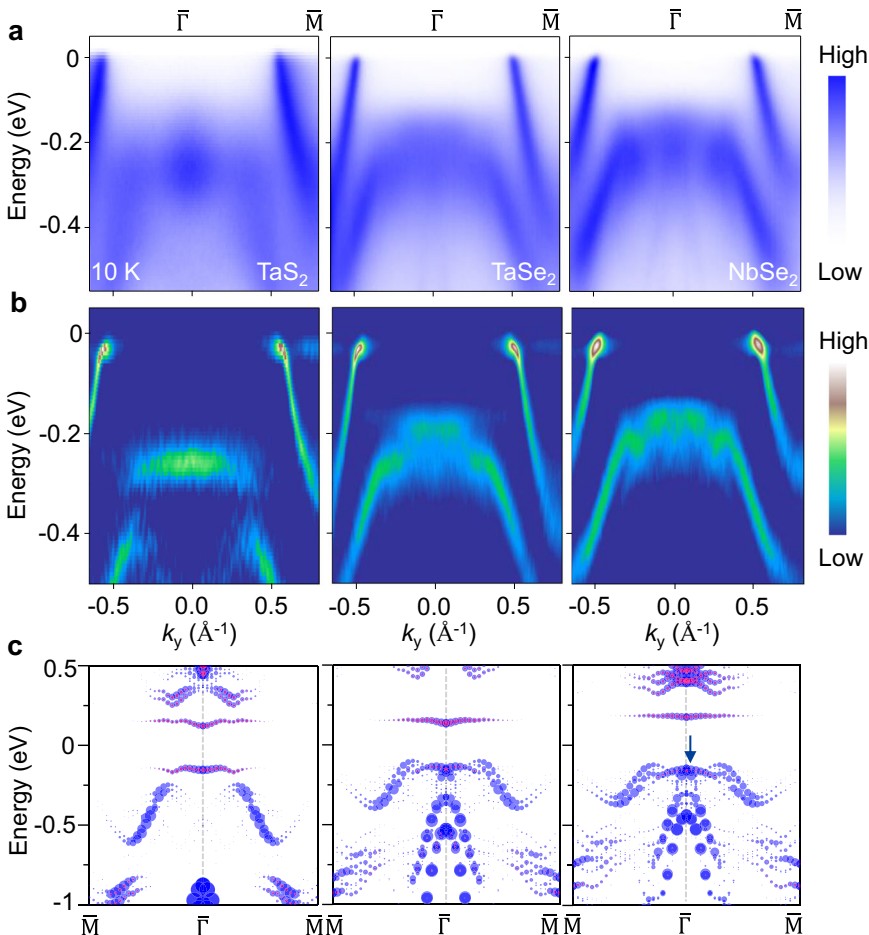

**Fig. 2 | ARPES spectra and orbital textures of monolayer NbSe$_2$, TaSe$_2$, and TaS$_2$ films. a** ARPES maps taken along the $\overline{\Gamma M}$ direction with the photon energy of 40 eV at 10 K. **b** Corresponding second derivative spectra for comparison. **c** Calculated unfolded band dispersions for these three monolayer materials with the ($\sqrt{13} \times \sqrt{13}$) superstructure. The dispersions for the projected $d_{3z^2-r^2}$ orbitals (red dots) of the central metal atom are superimposed on the top. The magnitude of the spectral weight of the $d_{3z^2-r^2}$ orbitals is amplified by 4 times for clarity. The blue arrow indicates the position of overlap between the LHB and the $p$ state.

As the CDW state and Mott phase occur together, the fit yields a CDW-Mott transition temperature of $T_C = 553 \pm 12$ K for monolayer 1T-NbSe$_2$. We performed the same types of measurements for other 1T phase TMDCs and obtained $T_C = 353$ K and 479 K for monolayer 1T-TaS$_2$ and 1T-TaSe$_2$[32]. The transition temperatures for monolayers are higher compared to the bulk counterparts[12–15], illustrating the robustness of the Mott gap in 2D 1T phase TMDC family. Interestingly, the Mott gap in monolayer 1T-NbSe$_2$ is the smallest among these three materials, but the derived transition temperature is the highest. As shown in Fig. 3e, the gap is reduced much slower with increasing temperature in 1T-NbSe$_2$ compared to the other two systems, and it becomes the largest at 330 K in NbSe$_2$. This scenario can be understood by a tunable bandwidth of the Hubbard band induced by varying hybridization strength between the chalcogenide band and the LHB in these materials.

**Orbital textures and *d-p* hybridization strength**

To investigate the effects of *d-p* band hybridization on the Hubbard band during the formation of $\sqrt{13} \times \sqrt{13}$ superstructure, we first study the orbital character of the Hubbard band structure and how the band is changed across the phase transition based on the first-principles results. The projected spectral weight of the Ta/Nb atoms in the (1×1) structure shows that $d_{3z^2-r^2}$ orbital dominates the conduction band crossing over the Fermi level (Supplementary Fig. 8). Similar dip structures in this conduction band are shown around the $\overline{\Gamma}$ point (Supplementary Fig. 7), which evolve into the dip structures of the

Hubbard band in the Mott phase (Supplementary Fig. 13), demonstrating the origin of the Hubbard band structure around the $\overline{\Gamma}$ point. As shown in Fig. 4a and Supplementary Fig. 7, the depth of the dip, which determines the bandwidth of the Hubbard band, changes from 124 meV in the (1×1) structure to ~10 meV in the $\sqrt{13} \times \sqrt{13}$ superstructure in monolayer 1T-NbSe$_2$, and the extracted bandwidth in monolayer 1T-TaS$_2$ in the $\sqrt{13} \times \sqrt{13}$ superstructure is ~28 meV, which is the largest among these three compounds. Experimentally, an analysis of integrated EDCs around the $\overline{\Gamma}$ point (±0.2 Å) indicates full width at half maximum (FWHM) in monolayer 1T-NbSe$_2$ is the smallest in these systems (Supplementary Fig. 9), consistent with the narrowest bandwidth in monolayer 1T-NbSe$_2$ predicted in the calculations. Note that it is difficult to accurately extract the bandwidth of the LHB from ARPES spectra as the broadening of the flat band from the hybridization.

Hybridization strength can be quantitatively evaluated from the spectral weight proportion of the $p$ states in the LHB. The overall hybridization strength in monolayer 1T-NbSe$_2$ is higher compared to the other two materials and the maximum hybridization is obtained at the position where the overlap occurs near the $\overline{\Gamma}$ point, as indicated by a blue arrow in Fig. 2c. It is ~50% in monolayer 1T-NbSe$_2$, 12% higher than the monolayer TaS$_2$ case. The calculated charge density distribution of the LHB shows orbital textures in the Mott phase and the relative size of the $p$ orbital is the largest in NbSe$_2$ among these three systems (Supplementary Fig. 10). As a simple approximation, we extract the value of Coulomb energy ($U$) from the spectral Mott gap

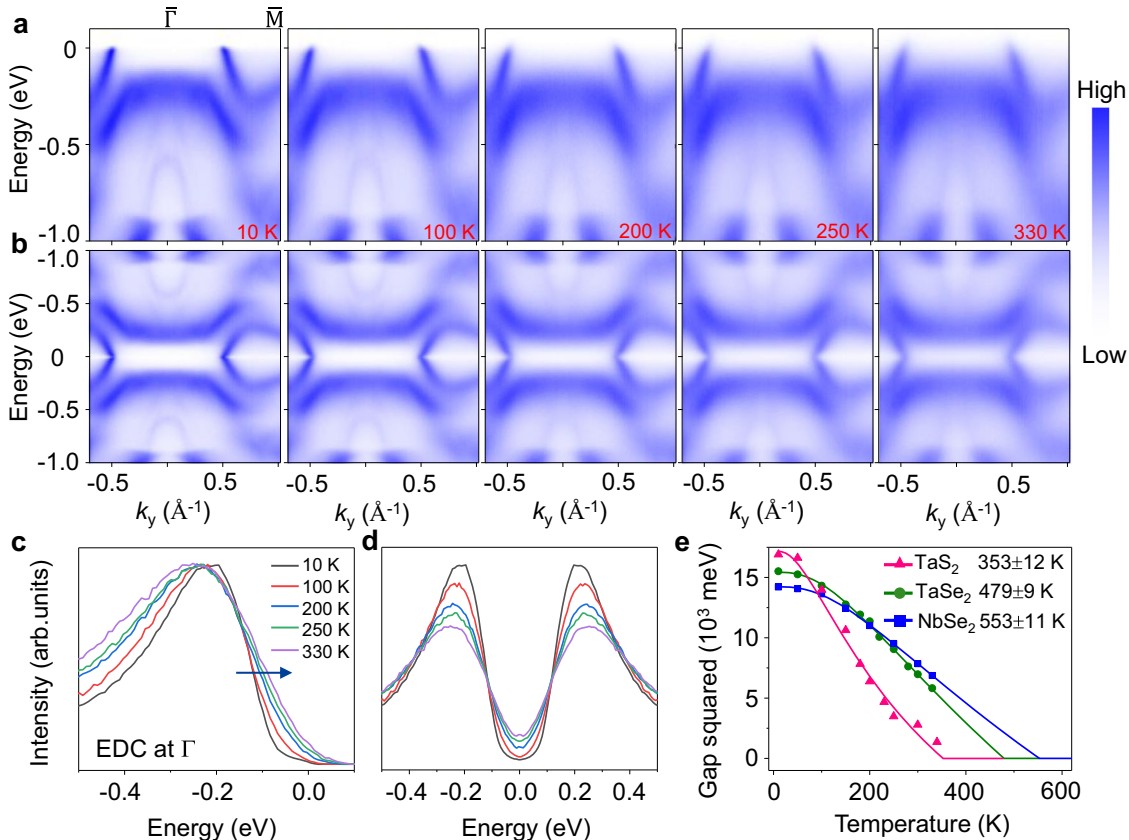

**Fig. 3 | Temperature dependence of the band structure and the Mott gaps.**
**a** Selected ARPES spectra of single-layer NbSe$_2$ taken along the $\overline{\Gamma M}$ direction
between 10 and 330 K. **b** Corresponding ARPES maps symmetrized in energy about
the Fermi level show a Mott gap which is gradually reduced with increasing tem-
perature. **c** Normalized EDCs and **d** unnormalized symmetrized EDCs at the zone
center at selected temperatures between 10 and 330 K show the evolution of the

gaps. The location of the leading-edge midpoints is indicated by a blue arrow. **e** The
extracted temperature dependence of the square of the Mott gap[32]. The solid
curves are fitting results using the BCS-type mean-field equation. The pink triangles,
green dots, and blue squares represent the results from TaS$_2$, TaSe$_2$, and NbSe$_2$,
respectively.

and obtain the bandwidth ($W$) of the Hubbard band from the calcu-
lated bandwidth of the metallic band in the CDW phase. As shown in
Fig. 4b, the value of interaction strength ($U/W$) increases from mono-
layer TaS$_2$ to NbSe$_2$ by ~2 times, following the trend of $T_C$, an effect
from the narrowing of the bandwidth that suppresses the inter-site
hopping of the electrons in monolayer NbSe$_2$. We plot them against
the $d$-$p$ band hybridization strength, verifying their correspondence
and indicating the $d$-$p$ band hybridization as an effective controlling
parameter to tune the bandwidth and lead to a more robust Mott
insulating ground state. It is shown that similar band hybridization may
have major effects on the electronic structure of high-temperature
cuprate superconductors and play an important role in controlling the
transition temperature[34–36]. A further computational study on the band
structure of monolayer 1T-NbSe$_2$ is carried out by imposing various Se
atomic displacements shifted along the z-axis to decrease the hybri-
dization strength of Nb and Se orbitals (Supplementary Fig. 11). The
resulting flat metallic band becomes broader at a larger displacement.
It can be understood that more Nb $d$ orbitals are involved in the for-
mation of Nb-Nb bonds as the Nb-Se bond becomes longer and results
in an increased Nb-Nb inter-site hopping at the smaller hybridization
strength.

Note that the phase transition is complicated, as a CDW transition
is also involved in these systems. We estimated the lattice distortion
(represented by the displacement of the transition metal atoms)
induced in the CDW phase. The value is obtained to be of 6% of the
lattice constant for monolayer TaS$_2$ and 8% for both the monolayer
TaSe$_2$ and NbSe$_2$. Therefore, the smaller value of the transition tem-
perature in monolayer TaS$_2$ is possibly partly due to a weaker CDW

instability, and the difference between TaSe$_2$ and NbSe$_2$ is dominated
by the varying hybridization strength. On the other hand, a weaker
CDW instability in monolayer 1T-TaS$_2$ combined with the on-site
Coulomb interaction, is supposed to result in a smaller opening energy
gap. However, the measured LEM gap is 11 meV larger in monolayer 1T-
TaS$_2$ compared to monolayer 1T-NbSe$_2$, indicating a difficulty of
opening a gap when there is a band hybridization, as it costs more
energy to hybridize the LHB with the chalcogen $p$ bands.

In summary, we provide ARPES measurements directly show the
detailed LHB structure in monolayer materials of 1T-MX$_2$ (M = Ta, Nb;
X = Se, S) family, together with the excellent agreement with our first-
principles calculations, establishing the tunable band structure and
thus $U/W$ in 1T transition metal dichalcogenides by hybridization
between the Hubbard band and chalcogenide band. Furthermore,
possible quantum spin liquid behavior is revealed by the reduced
intensity of the LHB and the closing of the energy gaps from the sur-
face doping by magnetic atoms (Supplementary Figs. 5 and 6). Our
study provides physical insights into the band coupling issue and
demonstrates a novel way for exploration of metal-insulator
transitions.

## Methods
### Film growth
Monolayer NbSe$_2$, TaSe$_2$, and TaS$_2$ thin films were grown on 4H-SiC
substrates in the integrated MBE/ARPES systems at the lab in Shanghai
Jiao Tong University. 4H-SiC was flash-annealed for multiple cycles to
form a well-ordered bilayer graphene on the surface. The films were
grown on top of the substrate by co-evaporating high-purity Nb/Ta and

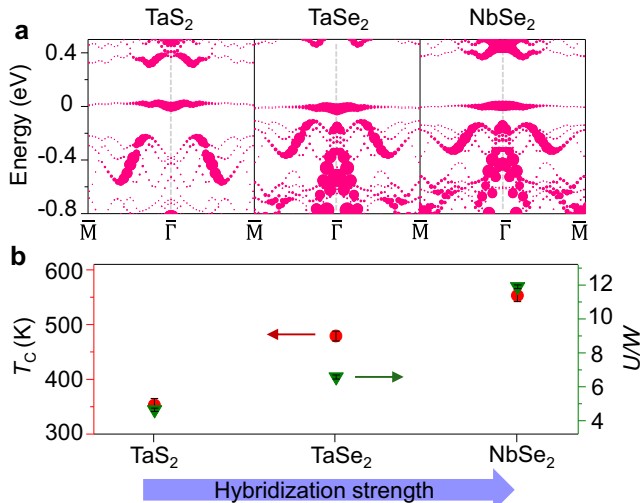

**Fig. 4 | Evolution of the band structure and transition temperature with *d-p* hybridization strength. a** Calculated unfolded band dispersions for monolayer 1T-TaS$_2$, 1T-TaSe$_2$, and 1T-NbSe$_2$ with the ($\sqrt{13} \times \sqrt{13}$) nonmagnetic superstructure, showing the suppressed bandwidth of Hubbard *d* bands with increasing *d-p* hybridization. **b** The extracted $T_C$ (red dots) for the three compounds and $U/W$ values (green inverted triangle) estimated from the Mott gap and bandwidth of metal band, plotted against the hybridization strength. Values of $U$ ($W$) are determined to be 130 (28), 125 (19), and 119 (10) meV for monolayer 1T-TaS$_2$, 1T-TaSe$_2$, and 1T-NbSe$_2$, respectively. Note that $U$ might be underestimated as the spectral gap is determined from LHB to the Fermi level. Error bars are estimated from the uncertainty of the fitting.

Se/S from an electron-beam evaporator and a Knudsen effusion cell, respectively. The optimized substrate temperature was set at 600, 650, and 600 °C for NbSe$_2$, TaSe$_2$, and TaS$_2$, respectively. The growth process and thickness of the films were monitored by RHEED, and the growth rate was set to 30 min per layer of film. The formation of metastable 1T phase TaSe$_2$ is favored at higher substrate temperatures but with poor film quality and 1H phase dominates below 500 °C, as shown in Supplementary Fig. 1. Co atoms were evaporated from electron beam evaporators onto the samples in MBE at room temperature and then transferred into the ARPES system in situ. K atoms were deposited using outgassed SAES Getter sources onto the samples kept at 10 K in the ARPES system.

### ARPES measurements
After growth, the films were transferred in situ to the ARPES system at the lab in Shanghai Jiao Tong University or transferred via a vacuum suitcase with a pressure better than $1 \times 10^{-10}$ mbar to the beamlines 03U and 09U at Shanghai Synchrotron Radiation Facility, in which the film surface was recovered through annealing at 300 °C before the experiments. ARPES measurements were performed at a base pressure of ~$5 \times 10^{-11}$ mbar with in-laboratory He discharge lamp (He-I 21.2 eV) and 30–100 eV photons at synchrotron using Scienta DA30 analyzers. Energy resolution is better than 15 meV and angular resolution is around 0.2°. Each sample's crystallographic orientation was precisely determined from the symmetry of constant-energy-contour ARPES maps. The Fermi level is determined by fitting ARPES spectra from a polycrystalline gold sample.

### Computational details from first principles
The DFT calculations were performed using the Vienna ab initio package (VASP)[37,38] with the projector augmented wave method[39] and the Perdew-Burke-Ernzerhof (PBE) functional in the generalized gradient approximation (GGA)[40,41]. Spin-orbit couplings were self-included in the calculations. A plane-wave energy cut-off of 300 eV and a 7 × 7 × 1 k-mesh were employed. Freestanding films were

modeled with a 15 Å vacuum gap between adjacent layers in the supercell. Hubbard $U$ is included through the simplified rotationally invariant approach for each Ta or Nb atom to account for the electron localization. Spin-orbit couplings are included in the calculations. The band structure and the Mott gap depend on the choice of $U$ and lattice constants. An excellent agreement with the ARPES spectra is obtained with the in-plane lattice constants (Hubbard $U$) of 3.56 Å (2.5 eV), 3.53 Å (2 eV), and 3.36 Å (2.27 eV) for monolayer 1T phase NbSe$_2$, TaSe$_2$, and TaS$_2$, respectively. The atomic structure modulated by the SOD CDW phase was fully relaxed until the force on each atom was less than 0.005 eVÅ$^{-1}$. These parameters are consistent with those reported previously[22,27,30,42]. We used the VASPKIT code for band-unfolding calculations[43].

### Data availability
General methods, experimental procedures, and characterizations are available within the article and the Supplementary Information. Other relevant data are available from the corresponding author upon request.

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

## Acknowledgements

We thank Prof. Yuanbo Zhang, Prof. Xiaoyan Xu, Dr. Xuejian Gao for helpful discussions. The work at Shanghai Jiao Tong University is supported by the Ministry of Science and Technology of China under Grant No. 2022YFA1402400 and No. 2021YFE0194100, the National Natural Science Foundation of China (Grant No. 12374188), the Science and Technology Commission of Shanghai Municipality under Grant No. 21JC1403000 (P.C.). Y.H.C. acknowledges support from the Ministry of Science and Technology, the National Center for Theoretical Sciences (Grant No. 110–2124-M-002-012), and the National Center for High-performance Computing in Taiwan. Part of this research is supported by Shanghai Municipal Science and Technology Major Project. P.C. thanks the sponsorship from Yangyang Development Fund.

## Author contributions

P.C. conceived the project. Q.G., H.Y.C., and P.C., with the aid of Z.H.C., Y.B.H., and Z.T.L., performed MBE growth, ARPES measurements, and data analysis. Y.H.C. and W.S.L. performed calculations. P.C., Q.G., and H.Y.C. interpreted the results. P.C. wrote the paper with input from other coauthors.

## Competing interests

The authors declare no competing interests.
