## [Transparent Peer Review file · Nature Communications]

Orbital textures and evolution of correlated insulating state in monolayer 1T phase transition metal dichalcogenides

Corresponding Author: Professor Peng Chen

Version 0:

Reviewer comments:

Reviewer #1

(Remarks to the Author)

The methodology and conclusion of the present work are quite similar to a previous NCOMM paper “Robust charge-density wave strengthened by electron correlations in monolayer 1T-TaSe2 and 1T-NbSe2” [Y. Nakata et al. (2021)], which the authors might overlook. ARPES and DFT study on monolayer 1T-TaS2 has also been published recently [Tian et al., Sci. China-Phys. Mech. Astron. 67, 256811 (2024)]. In general, the combination of electron correlations, a strong lattice distortion, and the absence of interlayer hopping is known to stabilize the Mott state. I do not see significant progress contained in the present work. Therefore, publishing the manuscript in NCOMM might be inappropriate.

Reviewer #2

(Remarks to the Author)

I co-reviewed this manuscript with one of the reviewers who provided the listed reports as part of the Nature Communications initiative to facilitate training in peer review and appropriate recognition for co-reviewers.

Reviewer #3

(Remarks to the Author)

The manuscript of Gao et al. addresses the Mott-like insulating state in a well-known class of transition metal dichalcogenides – TMD (NbSe2, TaS2, TaSe2) monolayers. While the topic might interest a wide audience, this reviewer believes that there are major faults in the experimental data and analysis which lead to misleading conclusions.

The major concern is the evident coexistence of 1H and 1T polytypes in all ARPES spectra. How and to what extent these mixed phases affect (or even harm) the conclusions is hardly predictable. Did the authors try to estimate the size of the 1T and 1H domains? Can they exclude the formation of 4Hb or 6R phases (that are natural 1T-1H heterostructures, especially in the Ta compounds)?

More specific comments follow:

line 35: the sentence “Transition metal compounds with unpaired d electrons in transition metal ions commonly exhibit Mott insulating states” is not true. This is not the only condition to be satisfied in order to observe a Mott transition.

line 47: the sentence “...the surface layer of the atoms deforms...” is not clear: in a bulk material all layers undergo the lattice distortion, not only the surface; while in a monolayer, what is the surface layer?

Fig.1c: authors should properly label the RHEED peaks and provide evidence also for the Ta compounds.

line 74 and Fig.2: “High resolution ARPES spectra of 1T...”: maybe authors refer to high “energy” resolution, but they cannot claim any spatial resolution since features of both 1T and 1H phases are present. What is the typical beam spot size on the sample? With such inhomogeneous monolayers, the author should have considered micro- or nano-ARPES experiments instead.

line 98 and Fig.3: it is not clear how the mean-field (BCS-type) argument can help clarify the nature of the Mott state. It applies quite well to the CDW phase transition (for instance, 479 K is very close to the commensurate-incommensurate transition temperature of TaSe₂). Authors fail to provide a proper relation/contrast between CDW and genuine Mott behaviors.

line 134 and Fig. 4: it is not clear how both U and W have been estimated (their values should be given). In particular W is normally the bandwidth of valence states: in a tight-binding approach it would be the overlap integral of d-electrons and therefore, referring to Fig.7a, it should span at least 1 eV. Thus, how can U/W be as high as 12 in Fig. 4?

In conclusion, the present state of the manuscript neither clarifies the Mott nature of 1T TMD, nor it adds a significant advance to the existing (and sometimes controversial) literature on Mott states in these compounds. Regrettably, I cannot recommend publication.

Version 1:

Reviewer comments:

Reviewer #1

(Remarks to the Author)

I have read the author's reply, which however does not change my original assessment of the manuscript.

Reviewer #2

(Remarks to the Author)

Reviewer #3

(Remarks to the Author)

The authors have revised the manuscript following my advice and some answers are satisfying. However, some major concerns regarding the main message of the manuscript still remain.

- Authors have clarified that the coexistence of 1H and 1T phases is due to separated domains and not to the formation of heterostructures (Fig. R1).

- I do not fully understand the reply (page 4) "Only surface layer was suggested to deform into SOD cluster in bulk 1T-TaSe₂", which is definitely incorrect.

- Concerning the reply to my question on the mean-field-like temperature behavior of the gap (page 5, related to Fig. 3e of the main manuscript), claiming that CDW and Mott phases are coupled is not sufficient to justify their observation. To me it seems that everything can be simply explained in terms of CDW, without the need to invoke Mott physics:

(i) DFT calculations in Fig.2 (revealing a rather good agreement with ARPES data), do not show the "localized LHB", but a dispersive band that originates from 5d Ta states, in agreement with the experimental evidence. Highlighting the $d_{(3z^2)}$ orbitals is misleading since several mixed Ta states are contributing (especially to the Ta band below E_F).

(ii) The temperature behavior in Fig.3e are perfectly justified within a CDW mechanism. I do not see any evidence of Mott physics. The gap opening might be simply the consequence of charge-ordering: charge accumulation in the central atoms of the SOD leads to state filling and thus larger binding energy of Ta states (forming the claimed LHB); on the other hand, charge depletion at the Ta atoms on the tips of the SOD leads empty Ta states that look like the HUB.

(iii) Concerning the ratio U/W and Fig 4: authors estimated U from ARPES experiment and used the computed W of the strongly localized state at E_F . However, the experimental evidence shows that W should be much larger since the measured Ta band is not flat (see Fig. 2a, 3a). The use of the calculated W from a specific band provides a large U/W ratio, supporting their conclusions, but is not fully/experimentally justified.

Side notes:

- (i) there is a considerable experimental effort - see extended data Fig.5-6 - on the effect of doping (magnetic and non-magnetic), but minor attention is given on the main manuscript. Maybe a separate manuscript should be devoted to doping.
- (ii) The difference between computations shown in Fig. 2c and 4a is not clear or properly addressed.

Overall, I still do not see the "strong" and "definitive" evidence of Mott physics in these 1T monolayers. The quality of data and analysis is certainly good enough for publication, but in my opinion it does not meet the high standards of Nat. Commun.

Version 2:

Reviewer comments:

Reviewer #3

(Remarks to the Author)

The authors have addressed all my concerns with additional computations, data analysis and arguments that I find convincing and that meet the standards of Nat. Commun. I still think that the data on the Co and K doping (Extended Fig. 5 and 6) do not truly help support authors' conclusions and they might be included in a separate publication.

Reviewer #4

(Remarks to the Author)

The manuscript by Gao et al. presents an ARPES studies of monolayer 1T NbSe₂, TaSe₂, and TaS₂. They observed the lower hubbard band and the Hubbard gap. They estimated the T_c for the CDW-Hubbard transition based on the temperature dependent gap. The following questions need to be addressed

1. Novelty/significance of the work is not clear. Similar ARPES measurements been done on bulk 1T TaS₂ (<https://www.nature.com/articles/s41467-020-18040-4>, <https://www.nature.com/articles/s41467-024-47728-0>), where the mott gap and lower hubbard band was observed. Is there anything strikingly different for the monolayer limit compared to bulk?
2. The grown films have a mixture of 1T and 2H, which both show up in the ARPES spectra. Can the authors try to show a pure 1T band structure? The authors can either reduce their ARPES beamspot size or to increase the 1T film domain size.

Reviewer #5

(Remarks to the Author)

Version 3:

Reviewer comments:

Reviewer #4

(Remarks to the Author)

The authors have address the comments. We recommend the paper for publication.

Reviewer #5

(Remarks to the Author)

Authors' point-by-point response to the reviewers' comments

Reviewers' comments:

Reviewer #1 (Remarks to the Author):

The methodology and conclusion of the present work are quite similar to a previous NCOMM paper “Robust charge-density wave strengthened by electron correlations in monolayer 1T-TaSe₂ and 1T-NbSe₂” [Y. Nakata et al. (2021)], which the authors might overlook. ARPES and DFT study on monolayer 1T-TaS₂ has also been published recently [Tian et al., Sci. China-Phys. Mech. Astron. 67, 256811 (2024)]. In general, the combination of electron correlations, a strong lattice distortion, and the absence of interlayer hopping is known to stabilize the Mott state. I do not see significant progress contained in the present work. Therefore, publishing the manuscript in NCOMM might be inappropriate.

Authors' response: We thank the reviewer for pointing out a previous paper about ARPES study on monolayer 1T-TaSe₂ and 1T-NbSe₂ [Nat. Commun. 12, 5873 (2021)]. We are aware of this work, which is a follow-up work of Ref. 21 [NPG Asia Materials 8, e321 (2016)]. We have included this work in the references.

The methods used in this work include molecular beam epitaxy growth of single-layer transition metal dichalcogenides films on bilayer graphene terminated SiC, angle-resolved photoemission characterization on the band structure of these films, and orbital-resolved first-principles calculations, which are commonly used and well accepted by the condensed matter community. Neither this work nor the above-mentioned paper intends to develop novel methods or improve the basic techniques.

We do not agree that our conclusions are quite similar to the previous paper. The main finding in this work is that the hybridization of the chalcogen p states with lower Hubbard band stabilizes the Mott phase via tuning of the bandwidth, whereas the previous paper focuses on the competition between the lattice distortion and interlayer/intralayer hopping. We found that the narrowing of the bandwidth leads to a more robust Mott insulating ground state as shown by a significantly higher T_C in monolayer 1T-NbSe₂, in which the hybridization strength is 12% higher compared to monolayer 1T-TaS₂.

A comparison shows that our high energy-resolution ARPES spectra are much sharper and cleaner, allowing us to extract the fine structures near the Fermi level, which is key to determine the lower Hubbard band (LHB) and the hybridization with the chalcogen bands. The LHB is identified by investigating the orbital textures of the calculated bands from first-principles and matching the ARPES spectra. Furthermore, the d - p band hybridization strength is extracted and shown to be an effective controlling parameter to narrow the bandwidth of the Hubbard band in these single-layer systems (Fig. 4).

These findings are not revealed in the previous studies.

We noticed the recent paper on 1T-TaS₂ [Sci. China-Phys. Mech. Astron. 67, 256811 (2024)] during the submission process, as also mentioned by the reviewer. This competing work only shows the overall band structure of single-layer 1T-TaS₂. The flat band near the Fermi level is even not clearly resolved, and the *d-p* hybridization strength is not extracted.

The effect of electron correlations, lattice distortion, and the interlayer/intralayer hopping can be complicated and we endeavor to reveal the effect of hybridization strength in a family of 2D Mott insulators. Our high-quality samples and high energy-resolution ARPES measurements enable the observation of tunable T_C with varying *d-p* band hybridization, which reveals a mechanism for tuning the Mott states beyond the traditional methods such as gating and doping. We hope the reviewer will agree.

Reviewer #2 (Remarks to the Author):

I co-reviewed this manuscript with one of the reviewers who provided the listed reports as part of the Nature Communications initiative to facilitate training in peer review and appropriate recognition for co-reviewers.

Authors' response: We thank the reviewer's time and effort. Please see the response to the reviewer who provided the report.

Reviewer #3 (Remarks to the Author):

The manuscript of Gao et al. addresses the Mott-like insulating state in a well-known class of transition metal dichalcogenides – TMD (NbSe₂, TaS₂, TaSe₂) monolayers. While the topic might interest a wide audience, this reviewer believes that there are major faults in the experimental data and analysis which lead to misleading conclusions.

Authors' response: We thank the reviewer for finding the topic might interest a wide audience. The constructive comments and suggestions given by the reviewer helped us improve the manuscript. We address the reviewer's concerns and specific technique questions in the following.

The major concern is the evident coexistence of 1H and 1T polytypes in all ARPES spectra. How and to what extent these mixed phases affect (or even harm) the conclusions is hardly predictable. Did the authors try to estimate the size of the 1T and 1H domains? Can they exclude the formation of 4Hb or 6R phases (that are natural 1T-1H heterostructures, especially in the Ta compounds)?

Authors' response: We understand the reviewer's concerns on the mixed phases of the films. The ratio between 1T and 1H phases can be tuned by controlling the temperature

and coverage of the sample during growth. The pure 1T phase single-layer TaSe₂ can be achieved at a growth temperature of ~700 °C (method section). As shown in the figure below, we compared the ARPES spectra of the mixed phase (1T + 1H) TaSe₂ and 1T phase TaSe₂. The extracted 1T phase band dispersions, and band gap are basically the same in two systems. The effect of 1H domains on 1T domains is very small if any. We perform most experiments on the mixed phase because higher growth temperature in pure 1T phase tends to generate more Se vacancies (defects). Sharper and cleaner ARPES spectra are obtained in the mixed phase as is evidenced in the smaller full width at half maximum of the peaks in the energy distribution curves shown in the panel d, which allow us to extract more accurately the band positions and gaps. The figure below has been updated in the Extended Data Fig. 1.

Fig. R1 ARPES spectra of monolayer TaSe₂ with varied growth temperatures. a-c, ARPES spectra taken along the $\bar{\Gamma}\bar{M}$ direction with He I α ($h\nu = 21.2$ eV) at 10 K on monolayer TaSe₂ grown at 500 °C (a), 650 °C (b), and 700 °C (c) show bands of 1H, 1H + 1T, and 1T phase, respectively. d, Extracted EDCs for 1H + 1T mixed phase and 1T phase at the $\bar{\Gamma}$ point. The peaks from H phase and T phase are labeled. e, The extracted band dispersions are plotted together for a comparison, showing the 1T phase bands are basically the same in the mixed phase and the pure phase.

The atomic topographic study on these systems have been done by previous scanning tunneling microscope (STM) measurements (Refs. 18-21). The domain size is

estimated to be tens of nm in diameter. There is indeed a small amount of 1T-1H heterostructure ($< 10\%$) is shown in the STM images. However, ARPES spectra did not reveal the band structure of these heterostructures which could be due to the very weak signals from these small islands and merge with the background. As an example, ARPES on $4H_b\text{-TaS}_2$ and $4H_b\text{-TaS}_x\text{Se}_{2-x}$ show bands around the $\bar{\Gamma}$ point at the Fermi level [npj Quantum Materials 9, 36 (2024); PRB 108, 115115 (2023)], which are not observed in our experiments.

More specific comments follow:

line 35: the sentence “Transition metal compounds with unpaired d electrons in transition metal ions commonly exhibit Mott insulating states” is not true. This is not the only condition to be satisfied in order to observe a Mott transition.

Authors’ response: We agree with the reviewer and weaken the point by rephrase the sentence to “Transition metal compounds with unpaired d electrons in transition metal ions can exhibit Mott insulating states”. This sentence has also been updated in the main text.

line 47: the sentence “...the surface layer of the atoms deforms...” is not clear: in a bulk material all layers undergo the lattice distortion, not only the surface; while in a monolayer, what is the surface layer?

Authors’ response: Only surface layer was suggested to deform into Star-of-David clusters in bulk $1T\text{-TaSe}_2$ due to the metal behavior in transport measurements and insulator behavior in STM results. However, we agree with the reviewer that all layers in $1T\text{-TaS}_2$ undergo the lattice distortion and rephrase the sentence to a more general statement “...the in-plane atoms deformed into ...” in the main text.

Fig.1c: authors should properly label the RHEED peaks and provide evidence also for the Ta compounds.

Authors’ response: Per the reviewer’s suggestion, we labeled the RHEED peaks for the compounds and the graphene underneath, as shown in the figure below. This figure has also been added as the Extended Data Fig. 12.

Fig. R2 RHEED patterns. a-c, RHEED patterns taken at room temperature for monolayer (a) NbSe_2 , (b) TaSe_2 , and (c) TaS_2 , respectively.

line 74 and Fig.2: “High resolution ARPES spectra of 1T...”: maybe authors refer to high “energy” resolution, but they cannot claim any spatial resolution since features of both 1T and 1H phases are present. What is the typical beam spot size on the sample? With such inhomogeneous monolayers, the author should have considered micro- or nano-ARPES experiments instead.

Authors’ response: The reviewer is quite correct that we mean the high energy-resolution, which has been updated in the main text.

The beam spot size is $\sim 30 \mu\text{m}$ in diameter in 03U beamline of Shanghai Synchrotron Radiation Facility (SSRF), but $\sim 700 \mu\text{m}$ in the lab of Shanghai Jiao Tong University.

As demonstrated above in the response to the main concern, the 1T phase bands are the same in the mixed phase and the pure phase. The beam spot size mainly affects the ratio of signal and the background noise.

line 98 and Fig.3: it is not clear how the mean-field (BCS-type) argument can help clarify the nature of the Mott state. It applies quite well to the CDW phase transition (for instance, 479 K is very close to the commensurate-incommensurate transition temperature of TaSe₂). Authors fail to provide a proper relation/contrast between CDW and genuine Mott behaviors.

Authors’ response: The CDW state and Mott phase are coupled to each other in these compounds. When the in-plane atoms are deformed into Star-of-David clusters with a $(\sqrt{13} \times \sqrt{13})$ periodicity, one unpaired electron is thus localized on the central Ta/Nb atom and forms a half-filled band in the system, and the insulating behavior is explained by the Mott-Hubbard mechanism. As shown in the figure below, the temperature dependent behavior of the intensity in the gap around the Fermi level is similar as the intensity trend in the CDW gap at $\sim -0.6 \text{ eV}$, suggesting the CDW and Mott transition occur together. As the band is blurry around the CDW gap at $\sim -0.6 \text{ eV}$, it is difficult to extract the accurate gap value as a function of temperature.

Fig. R3 **Temperature dependent intensity in the gaps.** (a), ARPES spectra taken at 10 K for monolayer TaSe₂. (b), Integrated intensity as a function of temperature. Regions of interest used for integrating the intensity were indicated as red and blue boxes shown in panel (a).

To clarify the relation between the CDW and Mott phases, we added in the text in the page 5 that “As the CDW state and Mott phase transition occur together, the fit yields a CDW-Mott transition temperature of T_C ...”.

line 134 and Fig. 4: it is not clear how both U and W have been estimated (their values should be given). In particular W is normally the bandwidth of valence states: in a tight-binding approach it would be the overlap integral of d-electrons and therefore, referring to Fig.7a, it should span at least 1 eV. Thus, how can U/W be as high as 12 in Fig. 4?

Authors’ response: In a simplified Hubbard model, W represents the inter-site hopping of the electrons and we define it as the width of the metallic band in the CDW phase (electron correlation is not included). Because the ($\sqrt{13} \times \sqrt{13}$) CDW with emergence of the flat band at the Fermi level is important for the formation of a Mott phase. If using the metallic band in the normal phase, the value of U/W will be much less than 1 which is unfavorable for the formation of a Mott phase. As shown in Fig. 4, the metallic band resides around the Fermi level and the width is determined to be 28, 19, and 10 meV for 1T-TaS₂, 1T-TaSe₂, and 1T-NbSe₂, respectively. U is extracted from the spectral Mott gap determined from leading edge midpoints and the values are extracted to be 130, 125, and 119 meV for 1T-TaS₂, 1T-TaSe₂, and 1T-NbSe₂, respectively. Note that U might be underestimated as the determined spectral gap is from LHB to the Fermi level. Large values of U/W indicate the robust Mott phase in these 2D systems. The definition of the band width is clearer in Supplementary Fig. 13 in [National Science Review 11, 144 (2024), <https://doi.org/10.1093/nsr/nwad144>], where the multilayer and bulk compounds are put together. Using this definition, it is estimated that the band width of bulk 1T-TaSe₂ is ~0.3 eV which is consistent with the literatures such as [PRL 90, 166401 (2003)] and [PRL 130, 156401 (2023)].

Per the reviewer’s request, we added the definition of the U and W in the text of page 6: “the value of Coulomb energy (U) from the spectral Mott gap and obtain the bandwidth (W) of the Hubbard band from the calculated bandwidth of the metallic band in the CDW phase”. We also put the values of U and W in the caption of Fig. 4. It reads “Values of U (W) are determined to be 130 (28), 125 (19), 119 (10) meV for monolayer 1T-TaS₂, 1T-TaSe₂, and 1T-NbSe₂, respectively. Note that U might be underestimated as the spectral gap is determined from LHB to the Fermi level.”.

In conclusion, the present state of the manuscript neither clarifies the Mott nature of 1T TMD, nor it adds a significant advance to the existing (and sometimes controversial) literature on Mott states in these compounds. Regrettably, I cannot recommend

publication.

Authors' response: Experimentally, a gap and the characteristic flat band around the Fermi level were observed in monolayer 1T phase compounds by ARPES measurements. The insulating behavior of the 1T phase compounds is explained by the Mott mechanism because the conventional band theory still predicts metallic band across the Fermi level. The lower Hubbard band is identified by investigating the orbital textures of the calculated bands from first-principles and matching the ARPES spectra. Furthermore, the d - p band hybridization strength is extracted and shown to be an effective controlling parameter to narrow the bandwidth of the Hubbard band in these single-layer systems, which has not been revealed by the previous studies.

Comparing with the previous reports on these compounds, our ARPES spectra are much sharper and cleaner, which becomes possible with the film growth of exceptional quality by a delicate control of the growth conditions and high energy-resolution synchrotron-based ARPES measurements. The results allow us to extract the gap and the fine structures more accurately. We believe that our results, based on finer measurements, are much more robust. We hope the reviewer will agree.

We sincerely hope that the explanations and revisions mentioned above help to address the reviewer's concerns.

List of major changes:

- ◆ We added the statement that CDW state and Mott phase occur together in the text.
- ◆ We added how the U and W are estimated in the text and their values in the caption of Fig. 4.
- ◆ Extended Data Fig. 1 has been updated by adding the comparison between the band dispersion of pure 1 T phase and mixed phase (1T + 1H) for monolayer TaSe₂.
- ◆ We have added the RHEED images on all the 2D compounds in Extended Data Fig. 10.
- ◆ We have added relevant references in the manuscript.

Authors' point-by-point response to the reviewers' comments

Reviewers' comments:

Reviewer #1 (Remarks to the Author):

I have read the author's reply, which however does not change my original assessment of the manuscript.

Authors' response: We thank the reviewer's time and effort.

Reviewer #2 (Remarks to the Author):

I co-reviewed this manuscript with one of the reviewers who provided the listed reports as part of the Nature Communications initiative to facilitate training in peer review and appropriate recognition for co-reviewers.

Authors' response: We thank the reviewer's time and effort.

Reviewer #3 (Remarks to the Author):

The authors have revised the manuscript following my advice and some answers are satisfying. However, some major concerns regarding the main message of the manuscript still remain.

Authors' response: We are pleased that the reviewer finds some answers are satisfying. We address the reviewer's remaining concerns in the following.

- Authors have clarified that the coexistence of 1H and 1T phases is due to separated domains and not to the formation of heterostructures (Fig. R1).

Authors' response: We thank the referee for agreeing with the clarification for the coexistence of 1H and 1T domains.

- I do not fully understand the reply (page 4) "Only surface layer was suggested to deform into SOD cluster in bulk 1T-TaSe2", which is definitely incorrect.

Authors' response: We apologize that the previous reply is not clear enough. The claim that "Only surface layer was suggested to deform into Star-of-David clusters in bulk 1T-TaSe2" was made in [National Science Review 11, 144 (2024)]. It is only a suggestion/assumption based on the metallic behavior in transport measurements and insulating behavior in STM results. However, it does not affect our results as we focus on the single layer system which is shown to be insulating in varied experiments including STM and ARPES (Refs. 20 and 21). As we also demonstrated in the previous

reply that we agree with the reviewer that all layers in 1T-TaS₂ undergo the lattice distortion and have rephrased the sentence.

- Concerning the reply to my question on the mean-field-like temperature behavior of the gap (page 5, related to Fig. 3e of the main manuscript), claiming that CDW and Mott phases are coupled is not sufficient to justify their observation. To me it seems that everything can be simply explained in terms of CDW, without the need to invoke Mott physics:

Authors' response: As there is an unpaired electron localized on the central metal atom in a star-of-David unit cell, conventional band theory involving CDW predicts a metallic band structure (Fig. 4), which contradicts with the ARPES results that there is a band gap around the Fermi level. Mott-Hubbard mechanism is a natural way and widely used to explain this ground state. We hope the reviewer will agree.

(i)DFT calculations in Fig.2 (revealing a rather good agreement with ARPES data), do not show the "localized LHB", but a dispersive band that originates from 5d Ta states, in agreement with the experimental evidence. Highlighting the $d_{(3z^2)}$ orbitals is misleading since several mixed Ta states are contributing (especially to the Ta band below E_F).

Authors' response: It is difficult to identify the LHB from the calculated results in Fig. 2 because the LHB is overlapped with other bands in single-layer 1T-TaSe₂ and 1T-NbSe₂. Single-layer 1T-TaS₂ is a clean case that the LHB does not overlap with other bands and exhibits as a flat band in both calculated results and ARPES spectra (Fig. 2). The Hubbard bands in TaSe₂ and NbSe₂ can be revealed from the orbital resolved bands in the $(\sqrt{13} \times \sqrt{13})$ nonmagnetic superstructure, as shown in the figure below. Without Hubbard U included, the Hubbard band structure is shown around the Fermi level as a flat band in these compounds. With U included in the calculations, the LHB is pushed away from the Fermi level and overlapped with other bands (Fig. 2), which makes it look like "dispersive".

We agree with the reviewer that $d_{3z^2-r^2}$ is not the only orbital contribute to the LHB, but it dominates in the spectral weight, as shown in the figure below. For the central metal atom, the spectral weight of $d_{3z^2-r^2}$ is ~90%-95% of the total spectral weight of the Hubbard band in the three compounds. Note that in the Fig. 2(c), only the spectral weight of $d_{3z^2-r^2}$ orbital of the central metal atom is shown as red dots, but spectral weight of all the metal atom orbitals is shown as blue dots. The figure below has been added as the Extended Data Figure 13.

Fig. R1 Calculated unfolded band dispersions for the $(\sqrt{13} \times \sqrt{13})$ superstructure of monolayer NbSe₂, TaSe₂, and TaS₂ films. (a)-(c), Calculated spectral weight projected on the five d orbitals from the metal atoms for the $(\sqrt{13} \times \sqrt{13})$ superstructure in the CDW phase.

(ii) The temperature behavior in Fig.3e are perfectly justified within a CDW mechanism. I do not see any evidence of Mott physics. The gap opening might be simply the consequence of charge-ordering: charge accumulation in the central atoms of the SOD leads to state filling and thus larger binding energy of Ta states (forming the claimed LHB); on the other hand, charge depletion at the Ta atoms on the tips of the SOD leads empty Ta states that look like the HUB.

Authors' response: The reviewer proposed a different charge ordering picture that the electrons are paired in the central metal atom and thus the corresponding band is fully filled. As there are total 13 unpaired electrons in a SOD unit cell, there will be 11 electrons distributed in the outside metal atoms of the SOD unit cell. Considering the odd number of unpaired electrons, it will lead to an unpaired electron and partially filled bands. In the conventional band theory, such an unpaired electron will result in a metallic band structure.

(iii) Concerning the ratio U/W and Fig 4: authors estimated U from ARPES experiment and used the computed W of the strongly localized state at E_F . However, the experimental evidence shows that W should be much larger since the measured Ta band is not flat (see Fig. 2a, 3a). The use of the calculated W from a specific band provides a large U/W ratio, supporting their conclusions, but is not fully/experimentally justified.

Authors' response: As demonstrated in the reply to the point (i), the LHB is overlapped

with other bands in single-layer 1T-TaSe₂ and 1T-NbSe₂ and the bandwidth can not be extracted actually from the ARPES spectra. However, we can extract the band width from ARPES of single-layer 1T-TaS₂ as its LHB is well separated from others. As shown in the figure below, the bandwidth is extracted as ~ 10 meV, which is comparable to the calculated result (28 meV). Therefore, use of W from the calculations will not lead to a very different U/W ratio from the experiments.

FIG. R2 Extraction of flat band peak positions of single-layer 1T-TaS₂. (a) ARPES spectra taken along $\bar{\Gamma}\bar{M}$ at 10 K. (b) EDCs extracted from the dashed box indicated in panel A. The red dots are the fitting results. (c) An example of fit to the EDC at the $\bar{\Gamma}$ point. A Lorentzian peak is used to represent the flat band peak. The red curve is the fitting result and the black dot curve is the Shirley background.

Side notes:

(i) there is a considerable experimental effort - see extended data Fig.5-6 - on the effect of doping (magnetic and non-magnetic), but minor attention is given on the main manuscript. Maybe a separate manuscript should be devoted to doping.

Authors' response: We would like to thank the reviewer's suggestion. As shown in Extended Data Fig. 5, the energy gap is reduced by surface doping with magnetic adatoms, a sign of in-gap states filling up the gap, which is attributed to the resonance states around the Hubbard band edge induced by the Kondo coupling between magnetic adatoms and the local moments of 1T-MX₂ (M = Ta, Nb; X = Se, S) (Ref. 42). Doping with nonmagnetic impurities does not reduce the gap, suggesting the chemical potential shift plays a major role in nonmagnetic impurity doping. These findings are consistent with the quantum spin liquid scenario in single-layer 1T-MX₂, which is a result from a Mott insulator when the quantum fluctuations are strong enough to suppress the ordering of the spins.

The physics of doping dependent behavior has been demonstrated in Refs. 41 and 42. Therefore, we put the data in the supplementary material to support the findings in the Mott insulating phase.

(ii) The difference between computations shown in Fig. 2c and 4a is not clear or properly addressed.

Authors' response: The main difference between computations in Fig. 2c and 4a is that the magnetic ordering and Hubbard U are involved in Fig. 2c, which is commonly used to model the electron correlation effect (Ref. 20). The calculation in Fig. 4a represents a conventional band theory which shows a metallic band structure. The DFT + U calculation in Fig. 2c is used to model the effect of the electron correlation and a ferromagnetism ground state reproduces the ARPES spectra very well. Per reviewer's suggestion, we have added in the texts that "A ferromagnetic ground state and Hubbard U are included in the calculations in Fig. 2c to reproduce the experimental results" to clarify the calculation method used in Fig. 2c and 4a.

Overall, I still do not see the "strong" and "definitive" evidence of Mott physics in these 1T monolayers. The quality of data and analysis is certainly good enough for publication, but in my opinion it does not meet the high standards of Nat. Commun.

Authors' response: Mott insulator is a system that is expected to be a metal by counting electrons in the unit cell and also from the conventional band theory, but it is an insulator due to the strong electron-electron interactions. This concept is proposed on a material based on both experiment and theory. There are several reasons why we resort to Mott insulator scenario.

1. We observed a band gap and a flat band around the Fermi level by ARPES measurements. In the single-layer 1T-TaSe₂ and 1T-NbSe₂, the flat band is overlapped with other bands and is identified with the help of the first-principles calculations.
2. Localized moment in 1T-TaS₂ CDW state has been supported by measuring the Kondo effect of 1T/1H TaS₂ heterostructure as the local moment can be exchange-coupled with the Ta *d*-band of the underlying 1H-TaS₂ layer forming a Kondo ground state (Ref. 18).
3. The conventional band theory predicts a metallic system. The electron correlation effect can be simulated using the DFT+U method. The Hubbard band is revealed by investigating the orbital textures of the calculated bands from first-principles and matching the ARPES spectra. The extracted interaction strength (U/W) using a simple Hubbard model is shown to be closely related with the phase transition temperature.
4. Our magnetic impurity doping dependent experiments supports the quantum spin liquid state in the system which is a result from a Mott insulator when the quantum fluctuations in the system are strong enough to suppress the ordering of the spins. The finding is consistent with the quantum spin liquid picture in previous reports on single-layer TaSe₂ (Refs. 41 and 42).

In light of these points, we believe the Mott physics is the best description of the results

and hope the referee will agree. To weaken the claim, we modified the title by changing “Mott” into “correlated”.

List of major changes:

- ◆ We changed the “Mott” into “correlated” in the title.
- ◆ The computation method used in Fig. 2c has been clarified in the texts.
- ◆ Calculated unfolded band dispersions for the ($\sqrt{13}\times\sqrt{13}$) superstructure have been added as Extended Data Fig. 13.

Authors' point-by-point response to the reviewers' comments

Reviewers' comments:

Reviewer #3 (Remarks to the Author):

The authors have addressed all my concerns with additional computations, data analysis and arguments that I find convincing and that meet the standards of Nat. Commun. I still think that the data on the Co and K doping (Extended Fig. 5 and 6) do not truly help support authors' conclusions and they might be included in a separate publication.

Authors' response: We appreciate the reviewer's acknowledgment that our data and analysis are convincing and meet the standards of Nature Communications.

We understand that the doping results in Extended Data Fig. 5 and 6 are not the primary focus of the study. However, these data are useful as they demonstrate that the magnetic atom doping dependent result is from the resonance states around the lower Hubbard band edge induced by the Kondo coupling between magnetic adatoms and the local moments of the compounds. This is an indirect evidence to support the existence of quantum spin liquid state in the system arising from a Mott insulator under strong quantum fluctuations. We would appreciate it if these data could be placed in the supplementary material, as it would complement the key findings regarding the orbital textures of the lower Hubbard band and the *d-p* hybridization effect in the Mott insulating phase.

Reviewer #4 (Remarks to the Author):

The manuscript by Gao et al. presents an ARPES studies of monolayer 1T NbSe₂, TaSe₂, and TaS₂. They observed the lower hubbard band and the Hubbard gap. They estimated the T_c for the CDW-Hubbard transition based on the temperature dependent gap. The following questions need to be addressed

1. Novelty/significance of the work is not clear. Similar ARPES measurements been done on bulk 1T TaS₂ (<https://www.nature.com/articles/s41467-020-18040-4>, <https://www.nature.com/articles/s41467-024-47728-0>), where the mott gap and lower hubbard band was observed. Is there anything strikingly different for the monolayer limit compared to bulk?

Authors' response: We thank the reviewer for pointing out two previous ARPES works on bulk 1T-TaS₂. We have added these references in the texts as Refs. 27 and 28. We also appreciate the reviewer's concerns and would like to take this opportunity to elaborate on the novelty/significance of this work.

First of all, low dimensionality reduces complexity of interlayer interactions arising from different stacking orders (Ref. 23, 24). Especially, collapse of Mottness is

observed for different stacking terminations in bulk 1T-TaS₂ (Ref. 24). Therefore, monolayer samples offer an ideal and clean 2D Mott system for investigating its intrinsic physical properties. We have successfully grown high-quality 2D samples and performed high energy-resolution APRES measurements, which enable the observation of tunable T_C with varying d - p band hybridization strengths.

Secondly, our results show that $T_C = 353$ K, 479 K, 553 K for monolayer 1T-TaS₂, 1T-TaSe₂ and 1T-NbSe₂, respectively, which are much higher compared to the bulk counterparts (Refs. 12-15), demonstrating the robustness of the Mott gap in 2D systems. Moreover, our study shows that T_C in monolayer 1T-NbSe₂ is ~ 1.6 times higher than the monolayer 1T-TaS₂, although the Mott gap is 11 meV smaller in monolayer 1T-NbSe₂ at 10 K.

Thirdly, to understand the above T_C - Δ anticorrelation behavior, we identified the lower Hubbard band by investigating the orbital textures of the calculated bands from first principles and matching the ARPES spectra. Importantly, an overlap between Hubbard band and Se p band is observed in monolayer 1T-NbSe₂ and 1T-TaSe₂, favorable for the formation of the band hybridization. We reveal that the d - p band hybridization strength can be an effective controlling parameter to narrow the bandwidth of the Hubbard band that suppresses the inter-site hopping of the electrons in these systems. The narrowing of the bandwidth leads to a more robust Mott insulating ground state as shown by a significantly higher T_C in monolayer 1T-NbSe₂, in which the hybridization strength is 12% higher compared to monolayer 1T-TaS₂.

Finally, our magnetic impurity doping dependent experiments supports the existence of quantum spin liquid state in the system arising from a Mott insulator when the quantum fluctuations in the system are strong enough to suppress the ordering of the spins.

This study demonstrates that T_C can be tuned through variations in d - p band hybridization, highlighting a potential mechanism for manipulating Mott states that complements traditional gating and doping techniques. We kindly ask the reviewer to take these significant findings into consideration.

2. The grown films have a mixture of 1T and 2H, which both show up in the ARPES spectra. Can the authors try to show a pure 1T band structure? The authors can either reduce their ARPES beamspot size or to increase the 1T film domain size.

Authors' response: We understand the reviewer's concerns regarding the mixed phases of the films. By carefully controlling the growth temperature and sample coverage, we can selectively grow different phases. For instance, the pure 1T phase single-layer TaSe₂ can be obtained at a growth temperature of ~ 700 °C (as detailed in the Method Section). The ARPES spectrum of a pure monolayer 1T phase TaSe₂ is shown in the panel c in the figure below. To address the impact of mixed phases, we compared the ARPES spectra of mixed-phase (1T + 1H) TaSe₂ and pure 1T-phase TaSe₂. The

extracted band dispersions and bandgap of the 1T phase are nearly identical in both systems, indicating that the influence of 1H domains on the 1T phase is minimal, if any.

We perform most experiments on the mixed phase because higher growth temperature required for the pure 1T phase tends to introduce more Se vacancies (defects). The mixed phase yields sharper and cleaner ARPES spectra, as evidenced in the smaller full width at half maximum of the peaks in the energy distribution curves (panel d), which allow us to extract more accurately the band positions and gaps. The figure below has been put in the Extended Data Fig. 1.

Fig. R1 ARPES spectra of monolayer TaSe₂ with varied growth temperatures. a-c, ARPES spectra taken along the $\bar{\Gamma}\bar{M}$ direction with He I α ($h\nu = 21.2$ eV) at 10 K on monolayer TaSe₂ grown at 500 °C (a), 650 °C (b), and 700 °C (c) show bands of 1H, 1H + 1T, and 1T phase, respectively. **d,** Extracted EDCs for 1H + 1T mixed phase and 1T phase at the $\bar{\Gamma}$ point. The peaks from H phase and T phase are labeled. **e,** The extracted band dispersions are plotted together for a comparison, showing the 1T phase bands are basically the same in the mixed phase and the pure phase.

The reviewer has suggested an intriguing approach involving the use of small-size beam spot to probe the small domains in ARPES measurements. Based on previous scanning tunneling microscope measurements (Refs. 18, 19), the domain size of the samples is estimated to be on the order of tens of nanometers in diameter. In this case, nanoARPES

might be appropriate for such a study. However, as we have demonstrated above, the influence of 1H domains on the 1T phase is negligible, suggesting the nanometer spatial resolution may not be necessary for our current study.

Reviewer #5 (Remarks to the Author):

Authors' response: We thank the reviewer's time and effort. Please see the response above.

Response to reviewers' comments

Reviewer #4 (Remarks to the Author):

The authors have address the comments. We recommend the paper for publiation.

Authors' response: We thank the referee for his/her time and effort and the positive recommendation.

Reviewer #5 (Remarks to the Author):

Authors' response: We thank the referee for his/her time and effort and the positive recommendation.